# Accounting for Underspecification in Statistical Claims of Model Superiority

## Abstract

Machine learning methods are increasingly applied in medical imaging, yet many reported improvements lack statistical robustness: recent works have highlighted that small but significant performance gains are highly likely to be false positives. However, these analyses do not take *underspecification* into account—the fact that models achieving similar validation scores may behave differently on unseen data due to random initialization or training dynamics. Here, we extend a recent statistical framework modeling false outperformance claims to include underspecification as an additional variance component. Our simulations demonstrate that even modest seed variability ($\sim 1\%$) substantially increases the evidence required to support superiority claims. Our findings underscore the need for explicit modeling of training variance when validating medical imaging systems.

## 1 Introduction

Machine learning is experiencing a reproducibility and validation crisis, and medical imaging is particularly affected [1, 2, 3]. Recently, Christodoulou et al. [4] estimated a high probability ($> 5\%$) of false outperformance claims in $86\%$ of classification and $53\%$ of segmentation papers.

However, this framework does not model *underspecification* [5]: models trained to similar validation accuracy can differ substantially out of distribution. In practice, this often appears as run-to-run variability across random seeds, leading to fluctuations in segmentation or classification scores [6, 7]. While averaging across seeds can stabilize estimates, it remains a source of uncertainty when comparing single models (rather than *distributions* of models): if statistically significant differences can occur across different seeds of the *same* model, what does it entail for the statistical comparison of *different* models?

**This work.** We extend the false-claim probability model of Christodoulou et al. [4] by introducing an underspecification term that captures seed-induced variance estimated from recent reproducibility studies. Through simulation, we quantify how this additional variance inflates the evidence threshold needed to claim outperformance using estimated magnitudes of underspecification from the literature [6, 7]. While preliminary, we hope that this work will help raise awareness about underspecification to the medical imaging community, and encourage its integration as a factor in model validation. Our code and is available at `https://anonymous.4open.science/r/underspecification_false_claims-7135/`

## 2 Methods

We provide a very brief overview of the Bayesian model used by Christodoulou et al. [4] to estimate the probability of false outperformance claims. Following their framework, a false claim occurs when the true performance ordering is reversed despite the observed ranking. Given two methods $A$ and $B$ with observed mean performances $\hat{\mu}_A$ and $\hat{\mu}_B$ (where $\hat{\mu}_A > \hat{\mu}_B$) and a testing set of size $n$, there is a concerning probability of a false claim if:

$$P(\text{false outperformance claim}) = P(\mu_A \leq \mu_B | \hat{\mu}_A, \hat{\mu}_B, n) \geq 0.05,$$

this means that there is a probability above $5\%$ that the true means $\mu_A$, $\mu_B$ have actually the reverse relationship than the one estimated empirically.

**Segmentation.** For segmentation using Dice Score Coefficient (DSC), the probability of false outperformance claim is defined as

$$P(\mu_A \leq \mu_B | \hat{\mu}_A, \hat{\mu}_B, n) = t_{n-1}\left(\frac{\hat{\mu}_B - \hat{\mu}_A}{SE_{AB}}\right), \quad SE_{AB}^2 = \frac{s_A^2 + s_B^2 - 2s_A s_B r_{AB}}{n} \tag{1}$$

where $n$ is the test set size, $s_A$, $s_B$ are the standard deviations of method $A$ and $B$ and $r_{AB}$ the model congruence (correlation between predictions), and $t_{n-1}$ is the quantile of the Student distribution with $n-1$ degrees of freedom. This model simulates a t-test comparing samples with means $\hat{\mu}_A$, $\hat{\mu}_B$ given the standard error $SE_{AB}$. To account for underspecification, we modify the standard error $SE_{AB,\text{underspec.}}^2 = SE_{AB}^2 + \delta_A^2 + \delta_B^2$. This additive term represents global variability induced by random seed initializations. This formulation assumes: **(1)** independence of seed effects across methods, justified by independent training with different random seeds, and **(2)** approximate normality of performance across seeds, supported by empirical observations [6, 7].

**Classification.** For classification, Christodoulou et al. [4] modeled the joint predictions of two classifiers as a $2\times2$ multinomial table with Dirichlet prior. As the derivation of this model is more involved, we refer the reader to the description in Christodoulou et al. [4]. Because only marginal accuracies are usually reported, they also made used of *model congruence* ($p_{11} = P(\text{both correct})$) to impute the off-diagonal counts, clamped to feasible bounds. Given the posterior Dirichlet distribution, the false outperformance probability is computed through Monte Carlo sampling. To account for underspecification, we model the reported accuracies as random variables: $\tilde{p}_A \sim \mathcal{N}(\hat{p}_A, \delta_A^2)$ and $\tilde{p}_B \sim \mathcal{N}(\hat{p}_B, \delta_B^2)$, where $\delta_A$ denotes the standard deviation due to seed variability.

**Model parameter estimation.** Christodoulou et al. [4] ] reported median model congruence values of $r_{AB} = 0.67$ (Q1: 0.44; Q3: 0.82) for segmentation and $p_{11} = 0.67$ (Q1: 0.47; Q3: 0.83) for classification. We used a grid search to estimate the values of $s = s_A = s_B$ for both models, and obtained $s_{\text{seg}} = 0.197$ and $s_{\text{clf}} = 0.737$.

To estimate underspecification variance, we leverage reproducibility studies that trained multiple models with different random seeds (Table 1). We set $\delta \approx \sigma_{\text{indiv}} = 0.01$ for both tasks, representing a median across observed variabilities (range: 0.002-0.024). This approximated the expected variability of a model for brain tumour or prostate segmentation using a single model. For classification, this approximated the variability observed in prostate cancer or 3D lymph node metastases classification using a single model, or pancreatic cancer classification using an ensemble.

# 3 Results

Our main results are presented on Figure 1. First, on the left column, we see our reproduction of the results of Christodoulou et al. [4], generally showing an agreement with their findings, even though some variability was observed at extreme values.

Our contribution is presented in the right column, where we see that the probability of false claims substantially increases even with a relatively minor variability introduced across methods. With a variability as little as $1\%$ across seeds, the threshold for confidently avoiding false claims is further

Table 1: Reported run-to-run standard deviations ($\sigma$) of performance metrics across random seeds in reproducibility studies, with corresponding dataset sizes.

| Task | Task / Dataset | $n_{\text{train}}$ | $n_{\text{test}}$ | $\sigma_{\text{indiv}}$ ($\sigma_{\text{ensemb}}$) |
|---|---|---|---|---|
| Segmentation Dice Score | Brain tumor [7, 8] | 387 | 97 | ~0.01 (N/A) |
| | Prostate [6, 9, 10] | 32 | 16 | 0.017 (0.006) |
| | Pancreas [6, 11] | 281 | 82 | 0.002 (0.001) |
| Classification AUROC | Prostate cancer [6, 12] | 417 | 157 | 0.010 (0.008) |
| | Pancreatic cancer [6, 11] | 537 | 188 | 0.022 (0.012) |
| | Lymph node metast. (2D) [6, 13] | 274 | 91 | 0.024 (0.005) |
| | Lymph node metast. (3D) [6, 13] | 274 | 91 | 0.012 (0.005) |

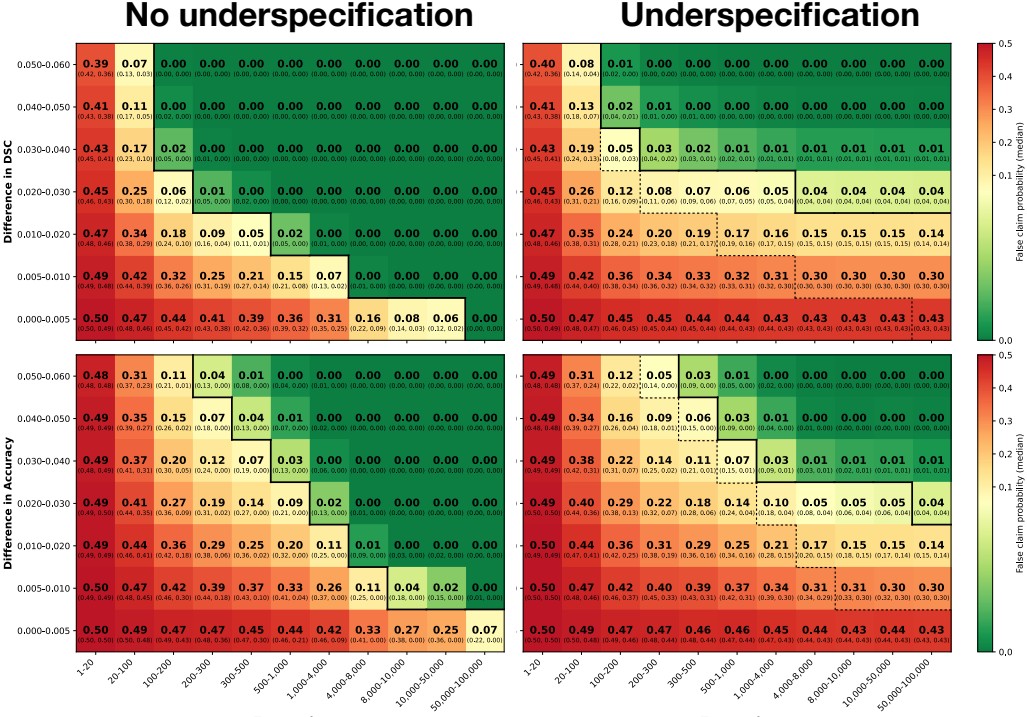

Figure 1: **Accounting for underspecification drastically increases the probability of false claims.** **(Top left)** Reproduction of the results of Christodoulou et al. [4] for segmentation. **(Top right)** Simulation of the impact of underspecification on segmentation using $\delta = 0.01$. The dashed line shows the trajectory of false claim probabilities above 0.05 without underspecification. **(Bottom left)** Reproduction of the results of Christodoulou et al. [4] for classification. **(Bottom right)** Simulation of the impact of underspecification on classification using $\delta = 0.01$.

raised to an extent where one requires large differences ($> 3\%$ in DSC or accuracy) to be confident that an outperformance claim is valid. Differences between methods in the order of $1\%$ are highly likely to yield false claims with an underspecification strength $\delta = 0.01$ indepedently of the size of the test set.

## 4 Discussion and conclusion

Our results show how underspecification affects false claim probabilities: it further raises the bar for being able to properly discriminate between methods. For classification, a variability across seeds of $1\%$ would mean that any testing set with fewer than 300 samples and differences across methods of $6\%$ Accuracy would have a probability of producing false claims above $5\%$. Similarly, for segmentation, testing sets below 100 samples should have differences above 0.04 DSC to achieve a low probability of false claims. Most existing papers fall below these differences or test set sizes and would then be subject to a high probability of false claims exacerbated by underspecification [2, 4].

A key limitation is that our simulations rely on variability estimates from datasets with only a few hundred samples (Table 1). Underspecification on larger test sets remains uncertain; we expect the overall variability to decrease, although subgroup-specific variability may remain substantial [5]. This means that our estimate of false claims might be overestimated on large testing sets. This is why we need a proper large-scale experimental validation to assess the extent of underspecification on medical imaging tasks.

Finally, while this work discussed underspecification as a variance on the global reported metrics, its effect is more evident when considering sub-groups (such as acquisition site, sex, age, etc.) [14, 15, 5]. We hypothesize that studying false claim probability not only at the global level but at the group level might reveal even more worrying trends and that many outperformance claims might not hold when *averaging performance across groups rather than globally*, where a good overall performance might hide poor systematic performance on some sub-groups [15]. These findings motivate more extensive stress testing of models across varied testing sets to better understand the extent of the problem caused by underspecification in medical imaging [16].

## Potential Negative Societal Impacts

While mostly positive societal impacts would stem from improving the validation AI models in healthcare, a potential risk would stem from a misuse of the results in this paper. This is a simulation study using estimated quantities, and thus should not serve as a basis for decision making, but as a call to further research on the topic.

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
