# OpenReview forum: "Accounting for Underspecification in Statistical Claims of Model Superiority"
_EurIPS.cc/2025/Workshop/MedEurIPS — EurIPS 2025 Workshop MedEurIPS Submission_

### Official Review · Reviewer_9kYV · 2025-10-29
**Nice study of how underspecification affects overclaiming and false positives**

**Rating:** 6
**Confidence:** 4

**Review:**

This paper extends a recent study on overclaiming in medical imaging, by adding in a term that takes underspecification into account -- namely the effect of models having the flexibility to agree in performance on the validation set while disagreeing substantially on out of distribution test data.

The paper makes an interesting theoretical derivation that ends in pointing out that this makes the issue of overclaiming even more urgent, as underspecification (which I presume is, if not the same, very close to overfitting) is a highly prevalent phenomenon. Underspecification is studied using methods from recent publications in machine learning.

There is a high dependency on models from previous works, making novelty a bit limited, but I still think the paper is highly interesting for discussion at medEurIPS.

---

### Decision · Program_Chairs · 2025-10-31

**Decision:**

Accept (Poster)

**Comment:**

The reviewer finds this an interesting and timely theoretical contribution that extends prior analyses of overclaiming in medical imaging by incorporating underspecification effects.